# Genotype distribution of human papillomavirus among women with cervical cancer stratified by HIV status in Tanzania

Alita Mrema[1,2‡], Mamsau Ngoma[1,2], Emmanuel L. Lugina [1,2‡]*, Atukuzwe Kahakwa[2], Chacha Josiah Mwita[1], Salama Iddy[1], Eulade Rugengamizi[3], Kandali Samwel[1], John Ngowi[1], Salum J. Lidenge[1,2‡], Charles Wood[4‡], Julius Mwaiselage[1,2‡]

1 Department of Clinical Research Training and Consultancy, Ocean Road Cancer Institute, Dar es salaam, Tanzania, 2 Department of Clinical Oncology, Muhimbili University of Health and Allied Sciences, Dar es Salaam, Tanzania, 3 Department of Oncology, Butaro Cancer Center, Burera, Rwanda, 4 Department of Interdisciplinary Medicine, Louisiana State University Health Sciences Center- New Orleans, Louisiana, United States of America

‡ AM and ELL share first authorship on this work. SJL, CW and JM are joint senior authors on this work
* elugina@yahoo.com

## Abstract

### Background

Cervical cancer (CC) is the leading cancer among women in Tanzania, especially among those between the ages of 15 and 44. The prevalence of high-risk Human papillomavirus (HR-HPV)-16/18 women in the general population at any given time is 3.3%. HR-HPVs 16 or 18 are the primary cause of CC. The distribution of HPV genotypes among women with CC according to HIV status is unknown in Tanzania. This study aimed to determine the HPV genotype distribution according to HIV status among women with CC in Tanzania.

### Methods

This cross-sectional study was done at Ocean Road Cancer Institute (ORCI) in Tanzania among women with histologically confirmed CC. HIV serology testing was performed. Biopsy was taken from cervical lesions, and DNA was extracted. HPV DNA was amplified by using a previously validated multiplex HPV PCR assay targeting 14 high-risk HPV genotypes (16,18,30,31,33, 35, 39, 45, 51, 52, 56, 58, 59, and 66) and two low-risk HPV genotypes (6 and 11). Continuous variables were compared using either a student t-test or the Mann-Whitney U test. Fisher's exact test was employed to compare discrete variables. A P-value less than 0.05 was considered statistically significant.

**Data availability statement:** All relevant data are within the manuscript and its Supporting Information files.

**Funding:** The authors are grateful for the seed funding program of the US National Institute of Health U54 CA190155 grant (C.W.), which supported this study. The funders of this study had no role in study design, data collection and analysis, decision to publish, or preparation of the manuscript.

**Competing interests:** The authors have declared that no competing interests exist.

**Abbreviations:** ART, Antiretroviral therapy; ADC, Adenocarcinoma; CC, Cervical cancer; CI, Confidence interval; HPV, Human papillomavirus; HR, High risk; ORCI, Ocean road cancer institute; WLWH, Women living with HIV

## Results

We included 100 women with CC. The prevalence of HIV infection in this study was 42%. The prevalence of any HPV infection was 94%, ranging from 1–3 genotypes per woman. HPV. The median age for women living with HIV (WLWH) with CC patients was 45 years (IQR, 31–60), while the median age for HIV-uninfected women with CC patients was 57 years (IQR, 30–78). (p = 0.0001). WLWH and HIV-uninfected women had similar HPV prevalence, except for HPV 35, which was more common in WLWH. There was a trend of high prevalence of HPV 52 and HPV 58 in WLHH compared to HIV-uninfected women, but this difference was not statistically significant. The prevalence of HPV 16 and/or 18 infection in the entire sample was 85%. The combined prevalence of HPV 16 and/or 18 was 76% WLWH and 91% amongst HIV-uninfected women (p = 0.036). The majority of women (77.9%) had single-genotype HPV infection. There was no difference in the distribution of multiple or single HPV genotypes infection by HIV status (p = 0.25).

## Conclusion

In this study, HIV positive women with CC presented at a significantly younger age (45 years) compared to the HIV-negative women (57 years). The prevalence of high-risk HPV is high among women with CC in Tanzania. Distribution of most high-risk HPV genotypes among women with CC was not significantly influenced by HIV status except for HPV 35, which appeared to be more in HIV positive women compared to HIV-negative women. While the majority of the high-risk HPV infections were with single HPV genotypes, the prevalence of multiple high-risk HPV infections was at 22%, with no significant difference between the two HIV statuses. A vaccination program that aptly targets HPV 16 and 18 could prevent up to 85% of CC cases in Tanzania, regardless of HIV. Keywords: Human papillomavirus, cervical cancer, HIV, Tanzania.

## Introduction

Cervical cancer (CC) is a significant worldwide public health concern, with it being the fourth most common cancer that affects women, behind breast, colorectal, and lung cancers, and the fourth leading cause of cancer deaths among women globally. One out of every 70 women develops CC in their lifetime [1].

Tanzania has the highest age-standardized incidence rate in East Africa, the third highest age-standardized incidence rate in Africa, and the sixth-highest mortality rate for CC in Africa [2]. The incidence rate for cervical cancer in Tanzania was 64.8 per 100,000 women, resulting in an estimated 10 868 new cases in 2023. An estimated 6,832 women died from the disease in that year, leading to an age-standardized mortality rate of 42.2 per 100,000 women [2]. It is predicted that without any intervention, a total of 341,955 women in Tanzania will die from the disease between 2020–2070, rising to 587,705 by 2120 [3]. Contributing factors to the high burden of CC in this

region include HIV endemic populations, unavailability of organized screening programs, and the use of visual inspection with acetic acid (VIA) as a primary screening method, which has variable test performance in resource-limited settings [4].

Most cases of CC are caused by persistent Human papillomavirus (HPV) infection [5]. There are thirteen high-risk (HR) HPV genotypes (16,18,31,33, 35, 39,45, 51, 52, 56, 58, 59, 66 & 68) classified as carcinogenic [5]. HPV genotypes 16 and 18 account for at least 70% of CC cases globally, with HPV 16 accounting for over 50% of all HR-HPV genotypes worldwide. It is postulated that HPV 16, in particular, may account for most cases of CC because of its innate ability to avoid immune surveillance [6]. HIV facilitates HPV acquisition and delays its clearance, resulting in persistent HPV infection with an increased risk of CC [7]. Moreover, other studies have reported that HIV modifies predilection towards multiple HPV genotype infections compared to the HIV-uninfected population [8].The estimated prevalence of any HPV infection in East Africa is among the highest globally (33.6%), including Tanzania [9] and 60% of CC cases are attributed to HR-HPVs 16 or 18 in Tanzania [10].

Adult HIV prevalence in Tanzania is estimated to be 4.9%. Approximately 6% of women with age 14–49 years are infected with HIV [11]. Due to the high burden of CC, Tanzania introduced the HPV quadrivalent vaccine in 2018, targeting 14-year-old girls. The vaccine is delivered through routine immunization, not campaigns or other point-in-time delivery strategies [12].

There is a lack of data on the HPV distribution among CC patients and how HIV affects the distribution of HPV genotypes in Tanzania. We hypothesized that HR-HPV types other than HPV16 and 18 are more frequently associated with cancer in WLWH versus HIV-uninfected women with CC. To address this hypothesis, we investigated the HPV genotype infection pattern among women with CC according to their HIV status. This information will help policymakers understand how the vaccination program can impact CC prevalence, particularly in HIV-endemic countries.

## Methodology

### Study design

This was a cross-sectional study done at Ocean Road Cancer Institute (ORCI) in Tanzania among women above 18 years of age with histologically confirmed CC. The ORCI is a tertiary public hospital in Dar es Salaam, Tanzania, annually attending over 1472 new CC cases. In this study, we enrolled 100 patients newly diagnosed with CC between November 2019 and January 2020. Simple random sampling was used to avoid selection bias. The ORCI research ethics committee reviewed and approved this study with an approval number 06/80/VOL.1. The study complied with the Helsinki Declaration of 2013. All participants provided written informed consent. The inclusion criteria were having histologically confirmed cervical cancer, known HIV status, and being over 18 years old. The exclusion criteria included refusing written informed consent and having previous malignancy.

All the participants received pre- and post-HIV test counseling. Participants were given proper counseling and treatment options according to the standard of care after disclosure of their HIV and biopsy results.

### Demographic, medical, sexual, and reproductive history

Social-demographic and clinical characteristics data was collected following the informed consent procedure using an interviewer-administered structured questionnaire.

### HIV testing and CD4 count

Rapid serum HIV antibody testing was done using the Alere Determine™ HIV Ag/Ab Combo kit (Inverness Medical, Massachusetts, USA). All positive cases were confirmed using the First Response® HIV1/HIV2 test kit (WB Premier Medical Corporation, Darman, India). A CD4 count was performed on all cases that tested positive for HIV with both kits. All CD4 tests were done according to the Omega Diagnostics package insert manufacturer's instructions.

## Histology

Following a thorough history and physical examination, a punch biopsy was taken from a cervical lesion was taken for confirmation of diagnosis. After fixation in neutral buffered formalin and processing of the biopsied tissue, formalin-fixed paraffin-embedded (FFPE) tissue blocks were made. The blocks were sectioned, 5-micron thickness slides were obtained, stained by hematoxylin and eosin (H&E)(Abbey Color, Philadelphia, USA), and examined by two trained pathologists to confirm or rule out CC diagnosis. One participant was excluded due to having two synchronous primary tumors. The final study sample consisted of 100 cases that met the inclusion criteria.

## DNA extraction and HPV genotyping

According to the manufacturer's instructions, DNA was extracted from the frozen tumor tissue utilizing the Qiagen DNeasy Blood and Tissue kit (Qiagen Inc., Valencia, CA, USA; catalog number 69506). The concentrations of the extracted genomic DNA were determined using a nanodrop spectrophotometer (Thermo Fisher Scientific, Wilmington, DE, USA). DNA specimens were stored at −20°C until PCR analysis.

A validated multiplex PCR kit was used to perform PCR reactions for 16 HPV genotypes (6,11,16,18,30,31,33, 35, 39, 45, 51, 52, 56, 58, 59, & 66). The PCR was performed in a single reaction tube using a type-specific Multiplex PCR kit (Qiagen Inc., Redwood City, CA, USA), following the manufacturer's instructions and as described previously [13,14]. For PCR amplification, a minimum of 50 ng of DNA sample solution (whether from a clinical sample or HPV plasmid DNA used as a positive control) was utilized as a template. The samples were subjected to incubation at 95°C for 15 minutes before undergoing 40 cycles of denaturation at 94°C (30 seconds), annealing at 70°C (90 seconds), and extension at 72°C (60 seconds). The analysis of PCR products was done using 6% polyacrylamide gel electrophoresis (PAGE) in 1XTBE and ethidium bromide staining. Gel images were captured with a ChemiDoc MP Imaging System (Bio-Rad; Hercules, CA). A positive genotyping result was determined if a clear band was visible on the gel. All HPV genotypes were detected by a single band, except for HPV types 16 and 58, which were detected by two separate bands [13,14].

## Statistical analysis

An Excel sheet was used for data entry. The study participants' descriptions were presented using percentages for discrete variables and mean (standard deviation) for continuous variables. Median and interquartile ranges were used where the continuous variables were not normally distributed. Shapiro-Wilk test was used to test for normality. The proportions and distribution of HPV genotypes were presented as percentages with 95% confidence intervals and histograms. The Mann- Whitney U test was used to compare independent groups when the dependent variable was not normally distributed and continuous. Fisher's exact test was used to compare independent groups when the dependent variable was discrete. Multiple HPV infections was defined as an infection with two or more HPV genotypes in an individual and single HPV infection was defined as infection with a one HPV genotype. A variable was considered significant if the p-value was < 0.05. All analyses were done employing IBM Corp. Released 2015. IBM SPSS Statistics for Windows, Version 23.0. Armonk, NY: IBM Corp.

## Results

We recruited 100 women with a diagnosis of CC in this study. In this study, 42% of participants had HIV infection. The prevalence of any HPV genotype was 94.

The median age of the all-study women was 50 years. The median age of WLWH and with CC was 45 years (IQR, 31–60), while the median age for HIV-uninfected women with CC was 57 years (IQR 30–78). (p = 0.0001). The use of alcohol was reported by over 50% of CC patients, but only 4% reported smoking. HIV status had an impact on the distribution of occupation status. Notably, the unemployment rate in the WLWH group was higher (79.66%) than that in the

HIV-uninfected group (51%)(p = 0.003). Generally, the majority of women (74%) were single. Most of the HIV-uninfected women were single (86.2%) compared to WLWH (57.1%)(p = 0.002). The overall median age of menarche was 15 years, while the median age of first sexual activity was 18 (Table 1).

About 77% of the women had an advanced-stage disease (stage III & IV). Of all women, 93% had squamous cell carcinoma (SCC), with just 7% having other histological types (adenocarcinoma, adenosquamous, and small cell carcinoma). Only 58% of women had information on their histological grade. The median duration of HIV diagnosis was seven months, but it ranged from 6 months to 20 years. The median CD4 count in WLWH was 507 cells/mm$^3$, and 97% of participants were on ART. The prevalence of HPV infection was 94%. Most participants had a single HPV genotype infection (77.9%). Among WLWH, the median CD4 count for HPV positive group was 521 cells/mm3, while that of HPV negative CC group was 453 (p = 0.67) (Table 2).

The prevalence of any HPV genotype was 94%. The number of HPV infections per woman ranged from 1 to 3 genotypes. The prevalence of HPV infection increased slightly from 91% in WLWH to 97% in HIV-uninfected women with CC. The difference was not statistically significant.

Eighty-five percent of women were infected with HPV 16 and/or 18. The combined prevalence of HPV 16 and/or 18 was 76% amongst WLWH and 91% amongst HIV-uninfected women (p = 0.036). There was a trend of high prevalence of HPV 52 and HPV 58 in WLHH compared to HIV-uninfected women, but this difference was not statistically significant. WLHA (11%) had a higher incidence of HPV 35 than HIV-uninfected women (0%) (p = 0.01) (Table 3).

Of the 14 high-risk HPV genotypes screened, 11 high-risk HPV genotypes (16,18,31,33, 35, 39, 45, 51, 52, 56, and 58) were identified. HPV 30, 59, and 66 were not detected in any tested samples. HPV 16 was the most detected genotype (47.6%), followed by HPV 18(16.8%) and HPV 33 (12.6%) (Fig 1).

**Table 1. The social and demographic characteristics of the study participants.**

| Characteristic | All Patients | HIV - | HIV + | P-value |
|---|---|---|---|---|
| | N(%) | N(%) | N(%) | |
| **Median Age (Years)(IQR)** | 50 (30-78) | 57 (30-78) | 45 (31-61) | 0.0001[a] |
| **Alcohol use** | | | | |
| Yes | 54 (54) | 32 (55.18) | 22 (52.39) | *1.000[b] |
| No | 46 (46) | 26 (44.82) | 20 (47.61) | |
| **Cigarette Smoking** | | | | |
| Yes | 4 (4) | 3 (5.18) | 1 (2.39) | *0.642[b] |
| No | 96 (96) | 55 (94.82) | 41 (97.61) | |
| **Education** | | | | |
| Primary | 65 (65) | 35 (60.23) | 30 (71.43) | 0.212[b] |
| Secondary | 35 (35) | 23 (39.77) | 12 (28.57) | |
| **Profession** | | | | |
| Employed | 32 (32) | 12 (20.34) | 20 (49.79) | 0.003[b] |
| Not Employed | 68 (68) | 46 (79.66) | 22 (51.21) | |
| **Marital Status** | | | | |
| Single | 74 (74) | 50 (86.20) | 24 (57.14) | 0.002[b] |
| Ever married | 26 (26) | 8 (14.80) | 18 (42.86) | |
| **Median Age at Menarche (Years)** | 15 (12–19) | 15 (12–19) | 14 (12–18) | 0.9479[a] |
| **Median age at first sexual intercourse in years (IQR)** | 18 (13–25) | 18 (13–25) | 18 (15–20) | 0.832[a] |

[a]-Mann- Whitney test; [b]-Fisher's exact test

**Table 2. The clinical characteristics of the study participants.**

| | All women | HIV - | HIV + | P-value |
|---|---|---|---|---|
| | N(%) | N(%) | N(%) | |
| **Cancer Stage** | | | | |
| Early (IB-IIA) | 23 (23) | 13 (22.41) | 10 (23.80) | 0.8339[b] |
| Advanced (IIB-IVA) | 77 (77) | 45 (77.59) | 32 (76.20) | |
| **Histology** | | | | |
| SCC | 93 (93) | 53 (91.4) | 40 (95.2) | 0.22[b] |
| Adenocarcinoma | 6 (6) | 5 (8.6) | 1 (2.4) | |
| Adenosquamous cell carcinoma | 1 (1) | 0 | 1(2.4) | |
| **Histological Grade** | | | | 0.263[b] |
| 1 | 19 (19.0) | 12 (20.69) | 7 (16.66) | |
| 2 | 30 (30.0) | 13 (22.42) | 17 (40.48) | |
| 3 | 9 (9.0) | 4 (6.89) | 5 (11.90) | |
| Missing | 42(42.0) | 29(50.0) | 13(30.96) | |
| **Median CD4 count (cell/mm$^{3)}$) (IQR)** | | | 507.98(17.30–1054.50)[a] | |
| **CD4 categories** | | | | |
| <200 | – | – | 8 (19.05) | |
| 200-499 | – | – | 12 (28.57) | |
| >500 | – | – | 22 (52.38) | |
| **ARVs status at baseline** | | | | |
| Yes | | | 41 (97.62 | |
| No | | | 1 (2.38) | |
| **HPV genotypes (n = 95)** | | | | |
| Single genotype | 74(77.9) | 45(81.8) | 28(71.8) | 0.25[b] |
| Multiple genotypes | 21(22.1) | 10(18.2) | 11(28.2) | |
| **Viral Load** | | | | |
| < 20 copies | | | 26(61.9) | |
| 21-100 copies | | | 8(19.1) | |
| 100-1000 copies | | | 3(7.2) | |
| >1000 | | | 1(2.3) | |
| Missing | | | 4(9.5) | |

[a]-Mann- Whitney U test; [b]-Fisher's exact test

Distribution of the HPV genotypes did not differ significantly between HIV positive and HIV-negative samples for HPV 16, 18, 31, 33, 39, 45, 51,52,58 and 56. However, HPV 35 was significantly more detected among HIV-positive samples compared to HIV-negative (p = 0.007) (Fig 2).

There was no difference in the distribution of multiple or single HPV genotypes infection by HIV status (p = 0.25) (Fig 3).

## Discussion

The prevalence of any HPV infection among women with CC in this study was 94%. This finding agrees with another study conducted in Zimbabwe that reported an overall HPV prevalence of 94% among women with CC [15]. Globally, there is a variation in the prevalence of any HPV genotype in CC, with Oceanian studies showing 88.3% and African studies showing 94.2% [4]. In this study, the prevalence of non-HPV CC was 6%, consistent with previous studies that have indicated that it ranges from 3–8% [16]. CC, which is not associated with HPV, is not well understood. The etiology is still elusive because

**Table 3. Percentage distribution of HPV infections according to HIV status of participants.**

| Characteristic | All Patients | HIV - | HIV + | P-value |
|---|---|---|---|---|
| | *Prevalence (95%CI)* | *Prevalence (95%CI)* | *Prevalence (95%CI)* | |
| **Any HPV infection** | | | | 0.31[a] |
| Positive | 94(87–98) | 97 (88–99) | 91 (77–97) | |
| Negative | 6 (2–13) | 3 (4–12) | 10 (3–27) | |
| **Type of infectionSi** | | | | 0.32[a] |
| Single | 78(68–86) | 83(70–91) | 72(55–85) | |
| Multiple | 22(14–32) | 18(9–30) | 28(15–50) | |
| **HPV 16** | | | | 0.13[a] |
| Positive | 68 (58–77) | 74 (61–85) | 60(43–74) | |
| Negative | 32 (23–41) | 26 (15–39) | 41(26–57) | |
| **HPV 18** | | | | 0.81[a] |
| Positive | 24 (16–35) | 22 (13–35) | 26 (14–42) | |
| Negative | 76 (66–84) | 78 (65–86) | 73 (58–61) | |
| **HPV 31** | | | | 0.72[a] |
| Positive | 8(4–15) | 7(2–17) | 9.5 (3–23) | |
| Negative | 92 (85–97) | 93(83–98) | 91(77–97) | |
| **HPV 33** | | | | 0.79[a] |
| Positive | 18(11–27) | 19(9–31) | 17(7–31) | |
| Negative | 82(73–89) | 81(69–90) | 83(67–93) | |
| **HPV 35** | | | | 0.01[a] |
| Positive | 5(2–11) | 0(0–6) | 11(4–26) | |
| Negative | 95(89–98) | 100(94–100) | 88 (74–99) | |
| **HPV 39** | | | | 0.57[a] |
| Positive | 3 (1–9) | 2 (0–9) | 5(1–16) | |
| Negative | 97(92–99) | 98(90–100) | 95(83–99) | |
| **HPV 45** | | | | 0.69[a] |
| Positive | 7(3–14) | 9(3–19) | 5(1–16) | |
| Negative | 93(86–97) | 91(81–97) | 95(83–99) | |
| **HPV 51** | | | | 0.66[a] |
| Positive | 2(0–7) | 2(0–9) | 2(0–13) | |
| Negative | 98(93–99) | 98(90–100) | 97(87–99) | |
| **HPV 52** | | | | 0.42[a] |
| Positive | 1(0–5) | 0(0–6) | 2(0–13) | |
| Negative | 99(94–100) | 100(93–100) | 97(84–99) | |
| **HPV 56** | | | | 0.39[a] |
| Positive | 5(2–11) | 7(2–17) | 2(0–13) | |
| Negative | 95(89–98) | 93(83–98) | 97(84–99) | |
| **HPV 58** | | | | 0.17[a] |
| Positive | 2(0–7) | 0(0–6) | 5(0–16) | |
| Negative | 98(93–99) | 100(93–100) | 95(83–99) | |

[a]-Fisher's exact test

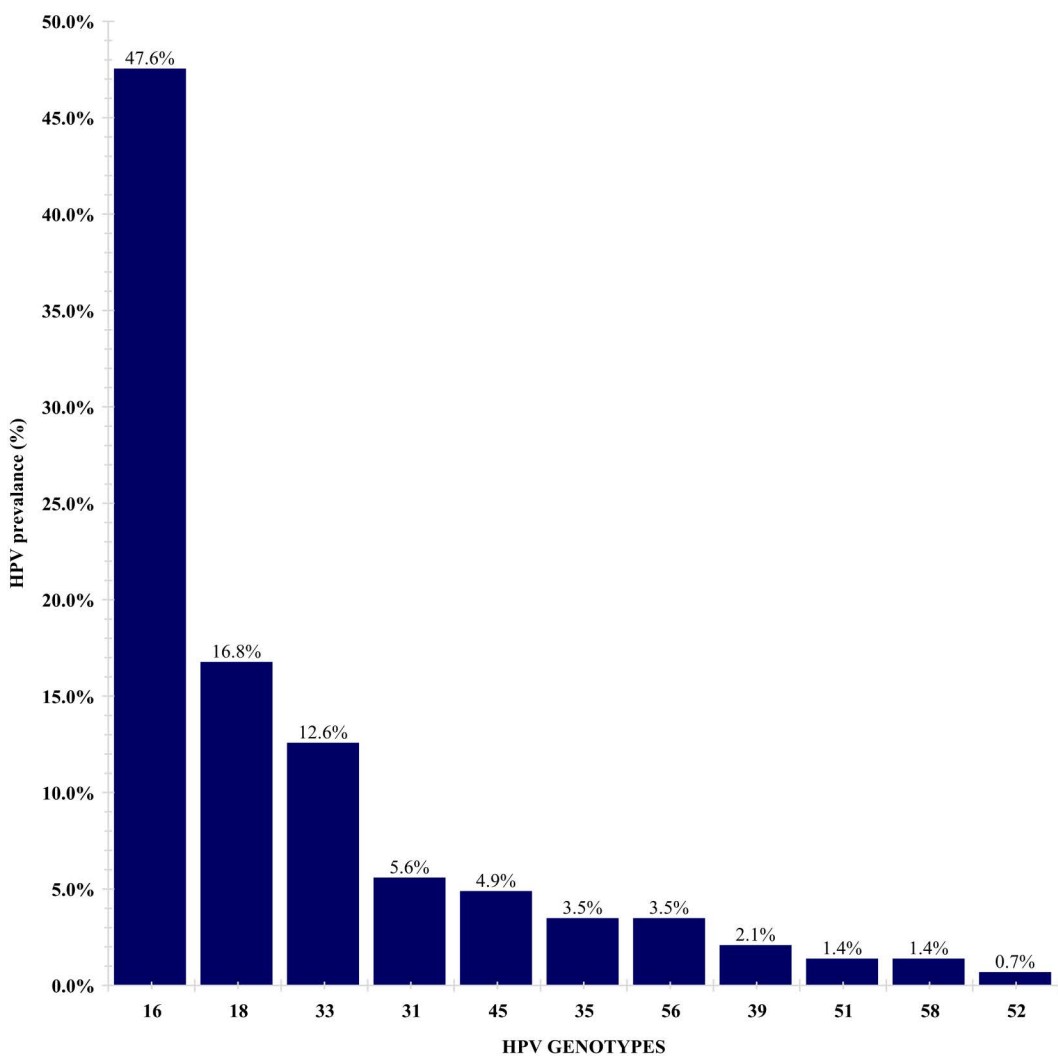

**Fig 1. Distribution of HPV genotypes.**

of limitations in research methodology, such as the absence of defined markers and model systems. Furthermore, HPV-negative CC research can be hindered by inaccurate diagnostic methods, leading to false HPV negativity [16].

The most common HPV genotypes to be isolated in women with CC regardless of HIV status were HPV 16 and 18, which were isolated in 85% of cases. The combined prevalence rates of HPV 16 and/or 18 were lower in WLWH (76%) than in HIV-uninfected women (91%) (p = 0.036). A high prevalence of HPV 16 and 18 among the HIV-uninfected women with CC compared to WLWH with CC has also been observed in Zimbabwe [15].

The HPV genotype spectrum observed in women with CC was distinct between those with WLWH and those who were not infected with HIV. There were more HIV-uninfected women with HPV 16, 33, 45, and 56 infections than those with WLWH. In contrast, WLWH had HPV 18, 35, 52, and 58 more frequently compared to HIV-uninfected women. However, only the prevalence of HPV 35 differed significantly between WLWH (12%) and HIV-uninfected women (0%) (p = 0.007). The prevalence of HPV genotypes in patients with CC seems to be influenced by HIV status. A study by Broker *et al.* that compared the distribution of HPV genotypes among women with end-stage renal failure according to their HIV status

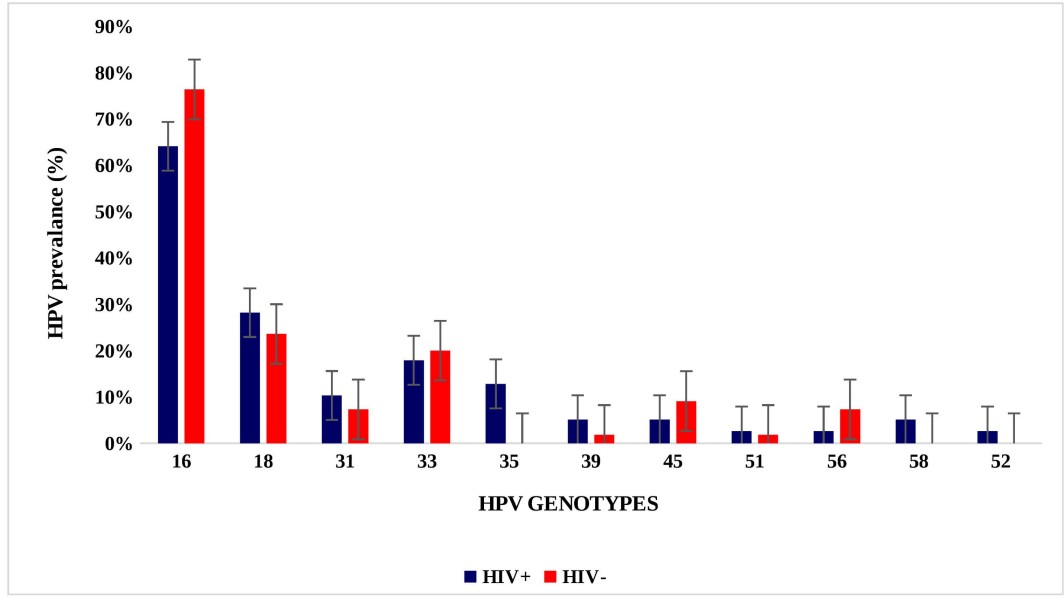

**Fig 2. HPV genotypes distribution by HIV status.**

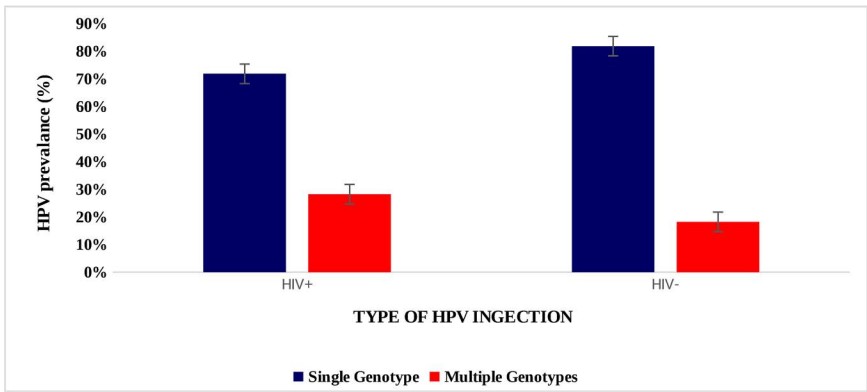

**Fig 3. Distribution of the number of HPV genotypes by HIV status.**

concluded that in addition to a wider variety and multiplicity of HR-HPV genotypes, WLWH also tend to be infected with non-carcinogenic HPV genotypes, or HPV genotypes with undetermined carcinogenicity like HPV-61, 62, 72, 81, 83 and 84 [7]. A meta-analysis of WLWH and CC in SSA showed that WLWH were less likely to be infected with HPV 16, but all other HR-HPV genotypes were more prevalent among WLWH [17]. A large sample size could detect the subtle difference in HR-HPV genotype distribution among women with CC based on their HIV status.

HPV 35 was more prevalent among WLWH than in HIV-uninfected women in the index study. This finding is consistent with the study by Mcharo *et al.* in southwestern Tanzania, which showed that HPV 35 was more prevalent among WLWH with high-grade squamous intra-epithelial neoplasia [18]. HPV 35 has also been found to be the second-most prevalent and persistent HPV infection in WLWH [19]. A systematic review conducted in 2020 among women in Ethiopia showed that there was a marked difference in the prevalence of high-risk HPV strains depending on HIV status, with HPV 35

accounting for 10% of infections in WLWH. Interestingly, a systematic review in SSA by Okoye *et al.* in 2021 regarding prevalent HPV genotypes in WLWH showed that WLHH with HPV 35 were twice as likely to develop CC compared to those with HPV 16 infections [20]. In all cases of HPV 35 infection in the index study, it was a single-genotype infection, emphasizing its carcinogenicity. A study by Pinheiro *et al.* suggested that there is a strong association between HPV 35 and CC carcinogenesis, particularly in women of African origin [21]. Although HPV 35 is linked to cervical dysplasia and CC in African women, there are few studies on its genomics and biology [22], and it is not clear how HIV interacts with HPV 35 to drive CC pathogenesis. Further studies are needed to investigate the role of HPV 35 in the pathogenesis of CC, particularly in this region where high prevalence has been reported.

In our study, we found that WLWH and HIV-uninfected women had different age distributions, with WLWH diagnosed with CC approximately 12 years earlier on average compared to HIV-uninfected women. This aligns with findings from other studies, which have reported that WLWH are diagnosed with CC at a significantly younger age compared to their HIV-negative counterparts [15,23].

In the index study, HPV 16 (47.6%), HPV 18 (16.8%), and HPV 33 (12.6%) were the most prevalent HPV genotypes, which mirror global patterns. A study by De Vuyst *et al.* reported that the HPV 16 and 18 are the most prevalent HPV genotypes in CC in SSA [7]. The third most prevalent HPV genotypes in CC after HPV 16 and HPV 18 were shown to be consistent in all world regions and over time, namely a combination of HPV 31, 33, 35, 45, 52, and 58, with rare exceptions [6]. These HR-HPV genotype distributions should be used in targeted vaccination and screening programs. Differences in HPV detection methods, geographical areas, and lifestyles may cause differences between our data and global data.

The index study revealed that 74% of women had a single HPV genotype infection. The distribution of multiple or single HPV genotype infections did not differ based on HIV status (p = 0.58). Similar findings were reported in Zimbabwe [15]. Our findings contradict the findings of a study in Tanzania, which reported a predominance of multiple HR-HPV-genotype infections in WLWH and CC [18]. While this could be attributed to our small sample size, recent studies have also shown that technological advances and better DNA extraction protocols have resulted in increased detection of HPV genotypes and, therefore, the unmasking of the role played by previously underestimated genotypes [15].

The median CD4 count was the same for both HPV-positive and HPV-negative women. The finding is consistent with studies that indicate that effective ART treatment does not significantly increase HR-HPV clearance, even if HIV viremia is suppressed [17]. Furthermore, rolling ART has not reduced the incidence of CC [11]. The persistence of HR-HPV genotype rates among WLHA on ART could be due to ongoing immune dysregulation, even with viral suppression.

This study's limitation was its small sample size. It also did not include enough patients with a diagnosis of adenocarcinoma to draw any definitive conclusions.

In conclusion, this study recapitulates the fact that CC is associated with high-risk HPV. Some high-risk HPV genotypes (35, 52, and 58) appear to be more associated with HIV infection among CC women in Tanzania. The younger age at CC diagnosis for WLWH signifies the impact of HIV on aging and increased susceptibility to cancer pathogenesis. The high prevalence of HPV 16 and 18 among CC women highlights the possible impact of a vaccine program that effectively targets HPV 16 and 18 that could prevent up to 85% of cervical cancer cases in Tanzania, regardless of HIV infection. Newer studies are needed to further investigate the role of HPV 35 in this population.

## Supporting information

**S1 Dataset.   SPSS file containing the data.**
(XLS)

## Acknowledgments

Our gratitude goes out to Happy Kaduri and Mary Matepo for their work in facilitating data capture.

## Author contributions

**Conceptualization:** Alita Mrema, Mamsau Ngoma, Charles Wood.

**Data curation:** Alita Mrema, Mamsau Ngoma.

**Formal analysis:** Emmanuel L. Lugina, Mamsau Ngoma, Atukuzwe Kahakwa, Chacha Josiah Mwita, Eulade Rugengamizi, Salama Iddy.

**Funding acquisition:** Alita Mrema, Charles Wood, Julius Mwaiselage.

**Methodology:** Emmanuel L. Lugina, Alita Mrema, Atukuzwe Kahakwa, Chacha Josiah Mwita, Eulade Rugengamizi, Kandali Samweli, Salama Iddy.

**Project administration:** John Ngowi, Charles Wood.

**Resources:** Emmanuel L. Lugina, Kandali Samweli, John Ngowi.

**Software:** Alita Mrema, John Ngowi.

**Supervision:** Alita Mrema, Salum J. Lidenge, Charles Wood, Julius Mwaiselage.

**Validation:** Emmanuel L. Lugina, Kandali Samweli, Salum J. Lidenge, Julius Mwaiselage.

**Visualization:** Emmanuel L. Lugina, Alita Mrema, Atukuzwe Kahakwa, Chacha Josiah Mwita, Eulade Rugengamizi, Charles Wood, Salama Iddy.

**Writing – original draft:** Emmanuel L. Lugina, Alita Mrema, Salum J. Lidenge.

**Writing – review & editing:** Emmanuel L. Lugina, Salum J. Lidenge, Charles Wood, Julius Mwaiselage.

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
