## [Decision Letter · Decision Letter 0]

Dear Dr. Lugina,

Thank you for submitting your manuscript to PLOS ONE. After careful consideration, we feel that it has merit but does not fully meet PLOS ONE’s publication criteria as it currently stands. Therefore, we invite you to submit a revised version of the manuscript that addresses the points raised during the review process.

We look forward to receiving your revised manuscript.

Kind regards,

Fengyi Jin, Ph.D.

Academic Editor

PLOS ONE

“The authors are grateful for the seed funding program of the US National Institute of Health U54 CA190155 grant (C.W.), which supported this study.”

Reviewers' comments:

Reviewer's Responses to Questions

**Comments to the Author**

1. Is the manuscript technically sound, and do the data support the conclusions?

Reviewer #1: Yes

Reviewer #2: Yes

2. Has the statistical analysis been performed appropriately and rigorously?

Reviewer #1: Yes

Reviewer #2: Yes

3. Have the authors made all data underlying the findings in their manuscript fully available?

Reviewer #1: Yes

Reviewer #2: No

4. Is the manuscript presented in an intelligible fashion and written in standard English?

Reviewer #1: Yes

Reviewer #2: Yes

Reviewer #1: The authors have a well written manuscript with clear results.

This manuscript however could be improved in some aspects. In your abstract, I suggest you rewrite your conclusion to include your key findings as you found other genotypes of HPV in your work.

In the main script under your methods describing histology, could you provide some more details on how this method was carried out.

Under HPV genotyping, you combined two methods, could you briefly described how you carried out this method to aid other investigators who want to use your method.

Under discussion, you highlight the involvement of HPV 35 differing in women living with HIV and those without HPV. could you explain why this occurs in your population of women in this study.

In your conclusion in the manuscript it appears your are discussing your findings. Please rewrite to give your key findings more clarity.

Reviewer #2: Congratulations on this commendable and well-executed study. I applaud the authors for their valuable contribution in investigating the distribution of HPV genotypes stratified by HIV status among women with cervical cancer in Tanzania—an area of significant public health importance. The manuscript is well-structured, and the study design and methodology are appropriately applied. However, upon thorough review, I have identified several areas that could benefit from improvement. Specifically, the language in certain sections could be refined to enhance clarity, precision, and scientific rigor. Additionally, some statements lack adequate supporting evidence or require further elaboration to ensure comprehensive understanding.

Title:

Consider: Genotype distribution of Human Papillomavirus among women with cervical cancer stratified by HIV status in Tanzania

Rationale

This title captures the study's focus on HPV genotype distribution, stratification by HIV status, and the geographical context, maintaining a scientific tone and precision

ABSTRACT

Methods:

“confirmed CC.HIV serology”

Please ensure a space is added between 'CC' and 'HIV.' Additionally, there are several instances throughout the manuscript where proper spacing is missing and should be corrected for consistency and readability.

“Fisher's exact test was employed to compare discrete variables.”

The Pearson’s Chi-squared test is used more frequently than the Fisher’s exact test in the tables within the results section. Consider specifying the use of the Pearson’s Chi-squared test instead of the Fisher’s exact test where applicable.

Introduction

The introduction section is brief, with limited and repetitive citations. It lacks critical information about cervical cancer screening programs in Tanzania, such as the availability and implementation of Pap smear screening—whether these are part of national priority programs or conducted opportunistically. Additionally, consider including details on histological aspects, such as the grades of premalignant lesions (CIN1, CIN2, CIN3) and the types of malignancies (e.g., squamous cell carcinoma and adenocarcinoma) in relation to HPV infection.

“There are thirteen high-risk (HR) HPV genotypes (16, 18, 31, 33, 35, 39,45, 51, 52, 56, 58, 59, &

68) classified as carcinogenic”

There are 14 high-risk (HR) carcinogenic HPV genotypes, but HPV 66 has been omitted.

“Tanzania has an estimated 18.8 million women over the age of 15 years who are at risk of

developing CC. Current estimates indicate that CC affects 10241 women per year, and 6525 of them die from it in Tanzania.”

This information appears to be taken from the ICO/IARC HPV Information Centre. My concern is that the information in this organization is not updated as they do not include the latest publications on HPV and HPV related lesions, they seem to change only the year of their published report. Consider including the latest publications with the estimates.

“About 3.3% of women in the general population are estimated to have HPV-16/18 infection at any given time, and 68.0% of CC are attributed to HR-HPVs 16 or 18”

I am unable to confirm the percentages presented based on the cited publication.

Please provide additional details about the three licensed HPV vaccines, including the HPV genotypes they target, and specify that Tanzania has opted for the quadrivalent HPV vaccine.

METHODOLOGY

HIV testing and CD4 count

What company and instrument was used to measure CD4 cell count?

Histology

Please provide additional details on the histological methodology prior to staining the slides with H&E, including the manufacturer's information for the H&E stain. Additionally, specify how many histopathologists were involved in screening the slides for quality control purposes.

Sample size

Kindly include a sample size determination to evaluate if the study was powered .

Statistical analysis

This section needs improvement, the STROBE guidelines will be immensely useful for

this project and your future projects.

The manuscript does not explicitly specify the data management tool utilized. It would

be helpful to clarify whether the data was collected and managed using Microsoft Excel

(specifying the version), Microsoft Access, Redcap, or any other specific data management system.

“The study participants' descriptions were presented using percentages for discrete variables and

mean (standard deviation) for continuous variables.”

The mean and standard deviation were used in the study’s results section.

“Median and interquartile ranges were used where the continuous variables were not normally distributed.”

Kindly indicate which normality test was used, either, Shapiro-Wilk test, Anderson-Darling test, and Kolmogorov-Smirnov test.

Usually the statistical tests are written in full, Mann-Whitney U test, Pearson’s Chi-squared test. Please include the Fisher’s exact test on this section as well.

“The Mann- Whitney and Chi-square tests were employed to evaluate associations.”

What associations are being evaluated?

Define both single and multiple HPV infections.

“The level of significance was set at p less than 0.05.”

Consider: A variable was considered significant if the p-value was < 0.05.

“All analyses were done employing SPSS version 23.”

Include the software’s full details including its city and country.

Ethics approval

Include an ethics approval section with an ethics clearance reference number.

RESULTS

Table 1 and Table 2

It is recommended to include a superscript (number or letter) alongside the p-values in the column. This superscript will indicate which statistical test was employed to calculate the respective p-values. Then show the statistical tests on the table footnote.

Table 2

Under the 'histology' variable, consider including the other types of malignancies.

Under the variable 'histological grade,' please specify that it refers to cervical intraepithelial neoplasia (CIN).

“Mean CD4 count (cell/mm3) (IQR)”

The word ‘mean’ is a typo, the correct word is median.

I would include a table that will show the prevalence of HPV infection according to HIV status. The outcome variable will be HIV status and the dependent variables will be:

Any HPV, single HPV infection, multiple HPV infection, HR-HPV infection, LR-HPV infection and probable HR-HPV.

I recommend including a 95% confidence interval (CI).

Rationale: Including a 95% CI provides a measure of precision for the estimate and conveys the range within which the true value is likely to fall. It enhances the interpretability and reliability of the results, allowing readers to assess the statistical significance and variability of the findings.

Figure 1, figure 2 and figure 3

On the y-axis of each figure, kindly indicate; HPV prevalence (%)

Figure 2 and figure 3

Include a statistical test that was used to obtain the p-value on the figure legend.

limitation

The authors discussed some limitation. However other limitations such as selection bias were not highlighted thus could influence the results.

**Do you want your identity to be public for this peer review?** For information about this choice, including consent withdrawal, please see our Privacy Policy

Reviewer #1: No

Reviewer #2: **Yes: ** Keletso Phohlo

---

## [Author Response · Author response to Decision Letter 1]

23 Feb 2025

I have addressed all the editors comments point by point

---

## [Decision Letter · Decision Letter 1]

GENOTYPE DISTRIBUTION OF HUMAN PAPILLOMAVIRUS AMONG WOMEN WITH CERVICAL CANCER STRATIFIED BY HIV STATUS IN TANZANIA

PONE-D-24-52953R1

Dear Dr. Lugina,

We’re pleased to inform you that your manuscript has been judged scientifically suitable for publication and will be formally accepted for publication once it meets all outstanding technical requirements.

Kind regards,

Fengyi Jin, Ph.D.

Academic Editor

PLOS ONE

Additional Editor Comments (optional):

Reviewers' comments:

Reviewer's Responses to Questions

**Comments to the Author**

Reviewer #1: All comments have been addressed

Reviewer #2: All comments have been addressed

2. Is the manuscript technically sound, and do the data support the conclusions?

Reviewer #1: Yes

Reviewer #2: Yes

3. Has the statistical analysis been performed appropriately and rigorously?

Reviewer #1: Yes

Reviewer #2: Yes

4. Have the authors made all data underlying the findings in their manuscript fully available?

Reviewer #1: Yes

Reviewer #2: Yes

5. Is the manuscript presented in an intelligible fashion and written in standard English?

Reviewer #1: Yes

Reviewer #2: Yes

Reviewer #1: I have read through your revised manuscript. You and your co authors have incorporated all the suggestions made in the initial review. Well done, your manuscript reads well.

Reviewer #2: (No Response)

**Do you want your identity to be public for this peer review?** For information about this choice, including consent withdrawal, please see our Privacy Policy

Reviewer #1: No

Reviewer #2: **Yes: ** Keletso Phohlo

---

## [Editor Report · Acceptance letter]

PONE-D-24-52953R1

PLOS ONE

Dear Dr. Lugina,

I'm pleased to inform you that your manuscript has been deemed suitable for publication in PLOS ONE. Congratulations! Your manuscript is now being handed over to our production team.

Kind regards,

on behalf of

Dr. Fengyi Jin

Academic Editor

PLOS ONE